# Immunohistological detection of small particles of *Echinococcus multilocularis* and *Echinococcus granulosus* in lymph nodes is associated with enlarged lymph nodes in alveolar and cystic echinococcosis

Johannes Grimm[1], Juliane Nell[1], Andreas Hillenbrand[2], Doris Henne-Bruns[2], Julian Schmidberger[3], Wolfgang Kratzer[3], Beate Gruener[4], Tilmann Graeter[5], Michael Reinehr[6], Achim Weber[6], Peter Deplazes[7], Peter Möller[1], Annika Beck[1], Thomas F. E. Barth[1]*

1 Institute of Pathology, University Ulm, Ulm, Germany, 2 Department of General and Visceral Surgery, University Hospital Ulm, Ulm, Germany, 3 Department of Internal Medicine I, University Hospital Ulm, Ulm, Germany, 4 Department of Internal Medicine III, University Hospital Ulm, Ulm, Germany, 5 Department of Diagnostic and Interventional Radiology, University Hospital Ulm, Ulm, Germany, 6 Department of Pathology and Molecular Pathology, University Zurich and University Hospital Zurich, Zurich, Switzerland, 7 Institute of Parasitology, University of Zurich, Zurich, Switzerland

* thomas.barth@uniklinik-ulm.de

## Abstract

### Background

Alveolar (AE) and cystic echinococcosis (CE) in humans are caused by the metacestode of the tapeworms *Echinococcus multilocularis* and *Echinococcus granulosus sensu lato (s.l.)*. Immunohistochemistry with the monoclonal antibodies (mAb) Em2G11, specific for AE, and the mAb EmG3, specific for AE and CE, is an important pillar of the histological diagnosis of these two infections. Our aim was to further evaluate mAb EmG3 in a diagnostic setting and to analyze in detail the localization, distribution, and impact of *small particles of Echinococcus multilocularis* (spems) and *small particles of Echinococcus granulosus s.l.* (spegs) on lymph nodes.

### Methodology/principal findings

We evaluated the mAb EmG3 in a cohort of formalin-fixed, paraffin embedded (FFPE) specimens of AE (n = 360) and CE (n = 178). These samples originated from 156 AE-patients and 77 CE-patients. mAb EmG3 showed a specific staining of the metacestode stadium of *E. multilocularis* and *E. granulosus s.l.* and had a higher sensitivity for spems than mAb Em2G11. Furthermore, we detected spegs in the surrounding host tissue and in almost all tested lymph nodes (39/41) of infected patients. 38/47 lymph nodes of AE showed a positive reaction for spems with mAb EmG3, whereas 29/47 tested positive when stained with mAb Em2G11. Spegs were detected in the germinal centers, co-located with CD23-positive

**Data Availability Statement:** All relevant data are within the manuscript and its Supporting Information files.

**Funding:** The author(s) received no specific funding for this work.

**Competing interests:** The authors have declared that no competing interests exist.

follicular dendritic cells, and were present in the sinuses. Likewise, lymph nodes with spems and spegs in AE and CE were significantly enlarged in size in comparison to the control group.

## Conclusions/significance

mAb EmG3 is specific for AE and CE and is a valuable tool in the histological diagnosis of echinococcosis. Based on the observed staining patterns, we hypothesize that the interaction between parasite and host is not restricted to the main lesion since spegs are detected in lymph nodes. Moreover, in AE the number of spems-affected lymph nodes is higher than previously assumed. The enlargement of lymph nodes with spems and spegs points to an immunological interaction with the small immunogenic particles (spems and spegs) of *Echinococcus spp*.

## Author summary

Echinococcosis is a life-threatening disease in humans that is caused by the larval stages of the tapeworms *Echinococcus multilocularis* and *Echinococcus granulosus s.l.*. For the histological diagnosis of alveolar echinococcosis (AE) and cystic echinococcosis (CE) the two monoclonal antibodies (mAb) Em2G11, specific for AE, and the mAb EmG3, specific for AE and CE are available. We have analyzed a large cohort of formalin-fixed, paraffin embedded human tissue (FFPE) specimens of AE (n = 360) and CE (n = 178) using these two antibodies. We show that the mAb EmG3 has a specific staining pattern of the metacestode stadium of *E. multilocularis* and *E. granulosus s.l.* while staining of the larval state of *E. multilocularis* is limited to the mAb Em2G11. We further identified so called "small particles of *Echinococcus multilocularis*" (spems) and "small particles of *Echinococcus granulosus*" (spegs) not only in the main lesion of AE and CE, but also in the adjacent tissue and in the vast majority of lymph nodes analyzed draining the main lesion. The lymph nodes with these microparticles are enlarged as compared to a control group. We conclude that immunochemistry with these two antibodies is a valuable adjunct for the final histological diagnosis of AE und CE. The frequent detection of microparticles of *E. multilocularis* and *E. granulosus* in lymph nodes of patients with AE and CE argue for a larger interface of interaction of the parasite with the host's immune system than previously assumed and point to further mechanisms of these infections in humans which may be the basis for immunological reactions in the host.

## Introduction

Extraintestinal infections of humans with tapeworms of *Echinococcus spp.* represent a worldwide disease burden [1,2]. The two main types of this disease are cystic echinococcosis (CE) and alveolar echinococcosis (AE) [3]. CE is caused by the larval stage of a complex of *Echinococcus* species or genotypes (*Echinococcus granulosus sensu lato (s.l.)*) and AE by a group of haplotypes of *Echinococcus multilocularis* [4,5]. CE is the most frequently encountered form of echinococcosis with worldwide prevalence and over one million infected humans [1,2]. Although AE has a lower incidence, the number of reported AE cases has been increasing since the turn of the millennium [6]. AE is restricted to the Northern hemisphere with high endemic regions in Central Europe, Northern and Central Asia, and Western China [1,6].

When untreated, AE has a worse prognosis than CE and is therefore considered to be one of the most life-threatening zoonosis in Europe [7].

Humans are infected as accidental dead-end hosts and are not involved in the perpetuation of the complex life cycle of the tapeworms. Most but not all human infections occur in the liver. In contrast to AE (over 95% primary lesions in the liver), CE may affect other organ systems such as the lungs [7,8]. The clinical diagnosis may be delayed in both diseases due to a long asymptomatic interval, and thus, the infection frequently is an accidental finding during sonography or computed tomography [7]. As seen in imaging and macroscopic analysis, the lesions of both AE and CE are characterized by metacestode proliferation. However, in CE mostly few typical cysts surrounded by the parasite laminated layer and the host tissue adventitial layer are seen, in contrast AE lesions manifest with an invasive growth pattern and multiple small alveolar structures in histological examination [9,10].

In humans both lesions are characterized by an acellular and glycoprotein-rich layer, called laminated layer [11], and a germinal layer, containing the vital cells of the larvae [12]. The perilesional reactive tissue consists of an inner zone of epithelioid cells, a fibrotic zone, and an outer rim of lymphocytes [13]. CE and AE are clearly distinguished histologically by the thickness of the laminated layer and a different morphology of accompanying necrosis and surrounding fibrosis [9].

In unclear liver lesions, cutting needle biopsies for histological diagnostics are the gold standard. Immunohistochemistry (IHC) is an important diagnostic tool supporting conventional histology, especially in small biopsy specimens not containing all the diagnostic histological features for AE and CE. Two primary antibodies against echinococcus antigens are currently available; these are the monoclonal antibody (mAb) Em2G11 and the mAb EmG3 [9,14,15]. mAb Em2G11 is specific for the mucin-type Em2 glycoprotein of *E. multilocularis* and recognizes AE in tissue or effusions containing the antigen [9,14–16]. For mAb EmG3, the antigen recognized has not yet been fully characterized, however, the antibody is *Echinococcus* genus specific and reacts with metacestodes of *E. multilocularis*, *E. vogeli* as well as with the genotypes G1, G4, G5, G6 and G7 of *E. granulosus s.l.* [9]. With IHC techniques using mAb Em2G11, the laminated layer is readily detected in human tissue infected by the larval state of *E. multilocularis*. Moreover, *small particles of Echinococcus multilocularis* (spems) have been detected in AE samples [13,14,17]. Most probably spems arise from shedding of the laminated layer [14,18]. Similar features have been described for the mAb EmG3. Using this monoclonal antibody on tissue sections infected by *E. granulosus s.l.*, small particles have been detected in the tissue of the main lesion and named *small particles of Echinococcus granulosus s.l.* (spegs) [9,14]. Spems and spegs are not limited to the main lesion but are also found in the host tissue surrounding the cystic lesions [9,14]. Furthermore, spems are located in the germinal centers and the sinuses of lymph nodes of AE patients [17,19].

The aim of this study was to evaluate the mAb EmG3 in a diagnostic setting using infected human tissue samples with manifest AE and CE and to define the localization and distribution of spems and spegs in human tissue. Furthermore, we analyzed the impact of spems and spegs on regional lymph nodes.

## Material and methods

### Ethics statement

Approval for this study was obtained from the local Ethics Committee of the University of Ulm (vote for usage of archived human material 03/2014) and was in compliance with the ethical principles of the WMA Declaration of Helsinki (see: Dtsch Arztebl. 2003: 1632); patient data were anonymized accordingly.

## Patient and tissue samples

In this study, we tested formalin-fixed paraffin-embedded (FFPE) tissue material of patients with the immunohistochemically approved diagnosis of AE (typical morphology and positive staining with mAb Em2G11) and CE (typical morphology and no staining with mAb Em2G11) following the criteria of Reinehr et al. [9]. The cohort for testing mAb EmG3 included 360 specimens of 156 patients with AE. Most samples originated from the liver (n = 240). Six samples were fluid specimens like pleural effusion (n = 1) and cystic fine needle aspirates (n = 5). Additional, 178 specimens of 77 CE patients were tested, including three fine needle aspirates. Both cohorts included bioptic material in addition to resected specimens (AE: n = 14; CE: n = 1). Furthermore, we analyzed eight samples that were suspicious of CE based on histology (hematoxylin and eosin (HE) staining and Periodic acid-Schiff (PAS) staining only). The sample characteristics are shown in Table 1.

For a more detailed analysis of regional lymph nodes, we used 95 individual lymph nodes of 25 AE patients. These samples originated from regional lymph nodes draining from the liver (n = 61), the gall bladder (n = 22) or other locations (n = 12). For CE, we included 41 individual regional lymph nodes from 12 patients (lung: n = 22; liver: n = 8; gall bladder: n = 6; other abdominal localization: n = 5). As the control group, 73 lymph nodes of eight individuals with no signs of AE or CE were analyzed (hilus lymph nodes of lung: n = 42; mesenteric lymph nodes from colon/rectum carcinoma: n = 28; cystic lymph nodes of gall bladder: n = 3). These lymph nodes were resected during routine surgery and showed no signs of metastasis or granulomas. For better comparison, we subdivided the cohorts of lymph nodes in abdominal and thoracic lymph nodes. In AE no thoracic lymph nodes were analyzed.

## Staining procedures

HE, PAS, and IHC staining were performed according to standard protocols [20,21].

For IHC, two different primary antibodies were used: the antibody mAb Em2G11 (IgG$_1$), which binds specific to the mucin-type Em2 glycoprotein of *E. multilocularis* and the newly

**Table 1. Topographic localization of samples of patients with alveolar (AE) or cystic echinococcosis (CE).**

|  | n of AE samples / n of AE patients | n of CE samples / n of CE patients | n of CE suspected samples / n of CE suspected patients |
|---|---|---|---|
| **n** | 360 / 156 | 178 / 77 | 18 / 8 |
| **abdominal cavity** | 10 (2.8%) /7 | 4 (2.3%) / 3 | 0 (0.0%) / 0 |
| **abdominal wall** | 2 (0.6%) / 1 | 3 (1.7%) / 2 | 0 (0.0%) / 0 |
| **appendix** | 0 (0.0%) / 0 | 1 (0.6%) / 1 | 0 (0.0%) / 0 |
| **bone** | 5 (1.4%) / 2 | 13 (7.3%) / 5 | 6 (33.3%) / 1 |
| **chest wall** | 4 (1.1%) / 3 | 0 (0.0%) / 0 | 0 (0.0%) / 0 |
| **colon** | 2 (0.6%) / 2 | 0 (0.0%) / 0 | 0 (0.0%) / 0 |
| **diaphragm** | 1 (0.3%) / 1 | 0 (0.0%) / 0 | 0 (0.0%) / 0 |
| **fluid samples** | 6 (1.7%) / 5 | 3 (1.7%) / 3 | 0 (0.0%) / 0 |
| **gall bladder** | 3 (0.8%) / 3 | 6 (3.4%) / 5 | 0 (0.0%) / 0 |
| **heart** | 3 (0.8%) / 1 | 1 (0.6%) / 1 | 0 (0.0%) / 0 |
| **liver** | 240 (66.7%) / 139 | 85 (47.8%) / 45 | 8 (44.4%) / 5 |
| **lung** | 5 (1.4%) / 4 | 18 (10.1%) / 9 | 3 (16.7%) / 1 |
| **lymph nodes** | 70 (19.4%) / 39 | 19 (10.7%) / 12 | 0 (0.0%) / 0 |
| **mamma** | 1 (0.3%) / 1 | 0 (0.0%) / 0 | 0 (0.0%) / 0 |
| **muscle** | 2 (0.6%) / 2 | 9 (5.1%) / 4 | 0 (0.0%) / 0 |
| **retroperitoneal** | 1 (0.3%) / 1 | 1 (0.6%) / 1 | 0 (0.0%) / 0 |
| **salivary gland** | 2 (0.6%) / 1 | 0 (0.0%) / 0 | 0 (0.0%) / 0 |
| **unknown** | 3 (0.8%) / 3 | 15 (8.4%) / 10 | 1 (5.6%) / 1 |

tested mAb EmG3 (IgM) which reacts with multiple *Echinococcus spp.*(*E. multilocularis*, *E. granulosus s.l.* (G1, G4, G5, G6, and G7), *E. vogeli*) [9,14]. The optimal dilution of antibodies was tested by previous titration experiments (Em2G11: 1:100; EmG3: 1:2000). Both antibodies were generated by Peter Deplazes (Institute of Parasitology, University Zurich, Zurich, Switzerland; for details see: [9,14,15]).

The immunofluorescence stainings were performed as a double stain with antibodies specific for CD23 (SP23, rabbit IgG, DCS Innovative Diagnostik-Systeme, Hamburg, Germany, 1:200) and the mAb EmG3. Antigen was retrieved using citrate buffer pH6 in a pressure cooker followed by treatment with T-EDTA Buffer pH9.0 in a steamer. As secondary antibody we used Cy3-conjugated AffiniPure Goat Anti-mouse IgM antibody (Jackson ImmunoResearch; West Grove, USA; dilution: 1:400) and polyclonal swine anti-rabbit immunoglobulins/ biotinylated (Agilent Technologies, Santa Clara, California, USA; dilution: 1:800). For the unconjugated alpha mouse bio antibody, Alexa Flour 488 Streptavidin (Thermo Fisher Scientific, Waltham, Massachusetts, USA; 1:1600) was used as fluorochrome. The counterstain was performed with DAPI (4′,6-diamidino-2-phenylindole). Control stainings with unspecific IgM isotype antibodies (Mouse IgM  Isotype control, BD Bioscience, Franklin Lake, USA, 1:2000) were performed according to the protocol above.

### Evaluation of stainings

The evaluation of the HE, PAS, and IHC stainings was carried out on multihead light microscope by four of the authors (JG, JN, AB, and TFEB) in a consensus-based fashion.

### Evaluation of lymph nodes

In pilot studies, we have described spems in lymph nodes [17]. In this study we analyzed the distribution of these particles in detail. We examined separately every individual lymph node on a slide for size, staining pattern, and staining intensity. To evaluate the size of the lymph nodes, we determined the area of a rectangle which enclosed the lymph node. The staining intensity was classified by four of us (JG, JN, AB, and TFEB) into the categories negative (-), weak positive ((+)), and strong positive (+) (S1 Fig). For measurements we used mAb EmG3-- stainings. WHO-PNM (parasitic mass in the liver, involvement of neighbouring organs, metastasis) staging was evaluated for AE patients [22]. Due to limitation of clinical data (e.g. missing ultrasound images) we could not provide the WHO Informal Working Group ultrasound classification for the CE cases [23]. Serological data for AE (Em2-ELISA) and CE (haemagglutination assay for echinococcosis spp.) were retrieved from the patients'record.

### Statistical analysis

The estimation of average value, the standard deviation, and the standard error of the mean of the lymph node evaluation were calculated for patient characteristics and metric measurements. The Fisher exact test was used to evaluate differences in frequency of positive staining results. For group comparison, we performed a two-sided t-test type 2 with equal variance using Excel (Microsoft Office 365). The result was regarded as significant for p-values $p < 0.05$.

## Results

### Patients

The cohort included 156 AE and 77 CE patients. The AE patients had a mean age of 52.7 years (median: 56 years, min: 12 years; max: 86 years). The AE cohort included 62 males, 93 females and one patient of unknown gender. The CE cohort had a mean age of 40.4 years (median: 42

**Table 2. Patient characteristics.**

|  |  | AE | CE | CE suspected | Control group |
|---|---|---|---|---|---|
| **n** |  | 156 | 77 | 8 |  |
| **gender** | male | 62 (39.7%) | 40 (51.9%) | 2 (25%) |  |
|  | female | 93 (59.6%) | 33 (42.9%) | 5 (62.5%) |  |
|  | unknown | 1 (0.6%) | 4 (5.2%) | 1 (12.5%) |  |
| **age** | mean | 52.75 years | 40.44 years | 49.25 years |  |
|  | median | 56 years | 42 years | 46.5 years |  |
|  | minimal | 12 years | 10 years | 29 years |  |
|  | maximal | 86 years | 73 years | 76 years |  |
| **lymph node cohort** |  |  |  |  |  |
| **n** |  | 25 | 12 |  | 8 |
| **gender** | male | 8 | 7 |  | 4 |
|  | female | 17 | 5 |  | 4 |
| **age** | mean | 37.7 years | 36 years |  | 45.5 years |
|  | median | 36 years | 32.5 years |  | 39 years |
|  | minimal | 18 years | 14 years |  | 37 years |
|  | maximal | 64 years | 66 years |  | 64 years |

years; min: 10 years; max: 73 years) and included 40 males, 33 females and 4 patients with unknown gender. Further patient characteristics are shown in Table 2.

## Evaluation of staining with mAb EmG3

As negative controls, mAb EmG3-immunohistochemistry on different non-infected tissue was performed (S1 Table and S2 Fig). Gallbladder epithelia, hepatocytes and the parasite *Trichuris suis* showed weak background staining. Further, we used IgM isotype controls to control specificity of mAb EmG3 staining. These showed no staining of echinococcus material, however, a likewise weak staining with lipofuscin-rich hepatocytes and *Trichuris suis* was revealed (S2 Table).

Next, we tested the antibody on our cohort of AE and CE specimens. The laminated layer in both species always appeared strongly positive. In contrast to absent staining with mAb Em2G11, the inner and outer layer of the protoscoleces showed a positive staining using mAb EmG3 (Fig 1D). Furthermore, we noted small particles with positive staining of a diameter of 2–20 μm. These particles have been previously defined as spems in AE and spegs in CE. The particles were located in the necrotic tissue adjacent to the laminated layer, in the border zone which represents the transition from necrosis to vital host tissue and in the surrounding vital host tissue. Likewise, blood vessels adjacent to the lesion partially contained spems and spegs. In liver tissue, spems and spegs were predominantly located in the sinuses. Additionally, we detected these positively stained particles in regional lymph nodes of AE and CE patients (AE: 52/70; CE: 19/19). Only three AE and three CE lymph node samples showed histological detectable fragments of the laminated layer as detected by conventional histology (HE and PAS). Moreover, we found spems in fluid samples like fine needle aspirates of cysts. In high magnification, we detected spems and spegs in macrophages and as floating particles in the fluid (Fig 2). We included 8 cases of echinococcosis which were previously classified as suspicious for echinococcosis. All 8 cases did not show an intact laminated layer or protoscoleces in the HE and PAS staining. There was no immunoreactivity in the staining with mAb Em2G11, therefore diagnosis of AE seemed to be unlikely. IHC with mAb EmG3 revealed positive staining of fragments of the laminated layer; in combination with the negative mAb Em2G11

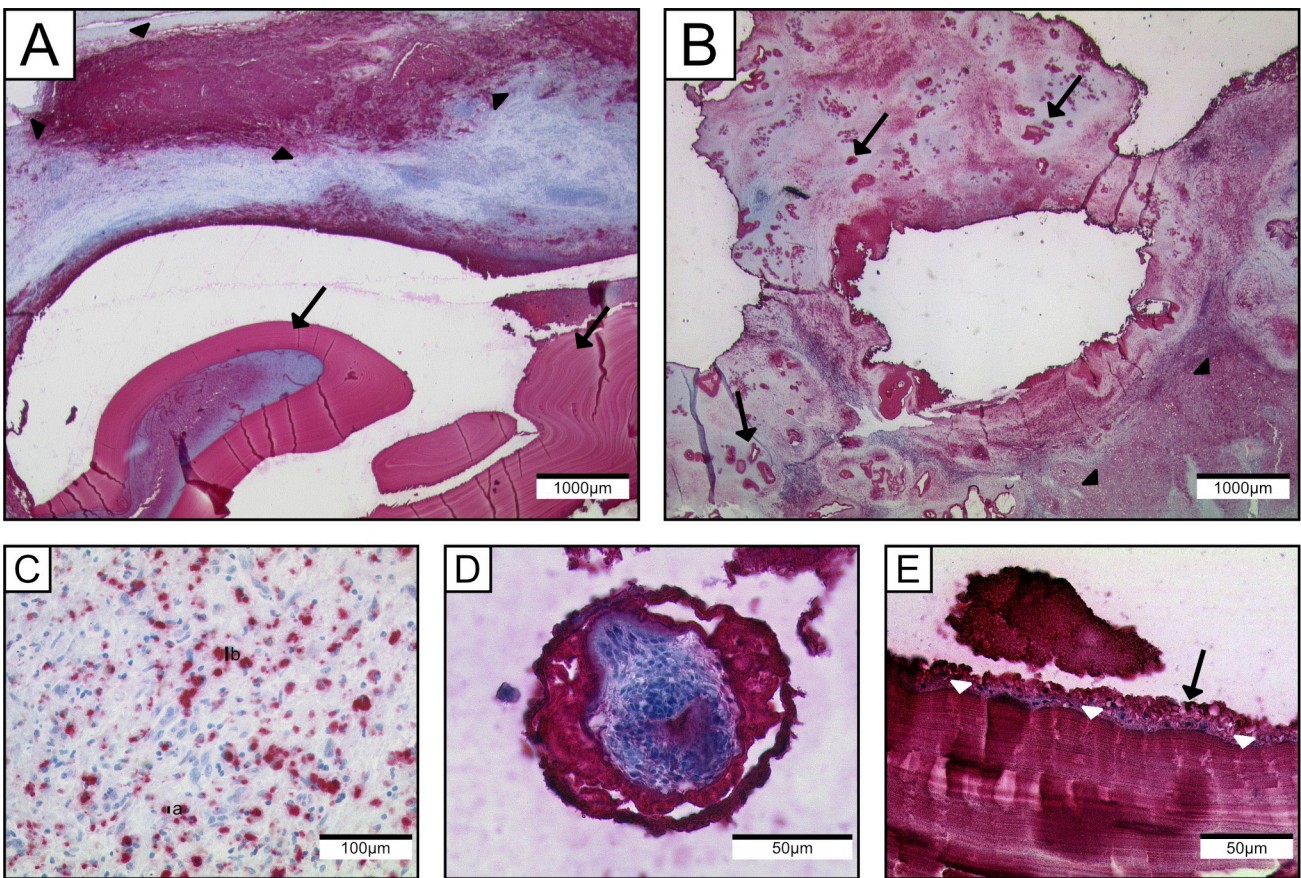

**Fig 1. Staining pattern of mAb EmG3 on echinococcus metacestodes.** mAb EmG3-IHC: (A) CE lesion in the lung showing strong positive staining of the laminated layer (arrows), adjacent to the main lesion small particles of *E.granulosus s.l.* (arrowheads) are strongly immunoreactive. (B) AE in the liver with strongly positive tubular structures presenting the laminated layer (arrows), adjacent to the main lesion small particles of *E. multilocularis* are strongly positive (arrowheads). (C) Higher magnification of spegs adjacent to the main lesion. Diameter of spegs: (a) 6μm (b) 17μm; (D) Protoscolex of *E. granulosus s.l.* with strong positivity in the periphery. (E) High magnification of laminated layer (white arrowheads) and germinal layer (black arrow) of *E. granulosus s.l.*

staining the diagnosis of CE was confirmed (diagnostic algorithm introduced in Reinehr et al. [9]). Staining patterns are given in Fig 1.

We noted that mAb EmG3 staining of spems was more intense than staining with mAb Em2G11. In 38 of 47 specimens of lymph nodes we detected spems, 9 more than tested with mAb Em2G11. Hence, mAb EmG3 staining has a higher sensitivity for spems in AE.

## Impact of spems/spegs on regional lymph nodes

Patient characteristics of the analyzed cohort are given in Table 2. In 39 of 41 (95.1%) lymph nodes of CE patients we detected spegs. In four of these lymph nodes (10.3%) we also detected fragments of the laminated layer. All 12 patients had at least one positive lymph node. In AE, we found spems in 63 of 95 (66.3%) analyzed lymph nodes; 8 (12.7%) of these were also infiltrated by laminated layers. 21 of 25 AE patients showed at least one positive lymph node. The number of affected lymph nodes in CE was significantly higher than in AE (Fisher's Exact Test: p = 0.0002). Neither in AE nor in CE, protoscoleces were detected in lymph nodes.

Analysis of the distribution pattern of mAb EmG3-positive particles (spems/spegs) in lymph nodes showed a similar pattern in both AE and CE with spems and spegs located in the

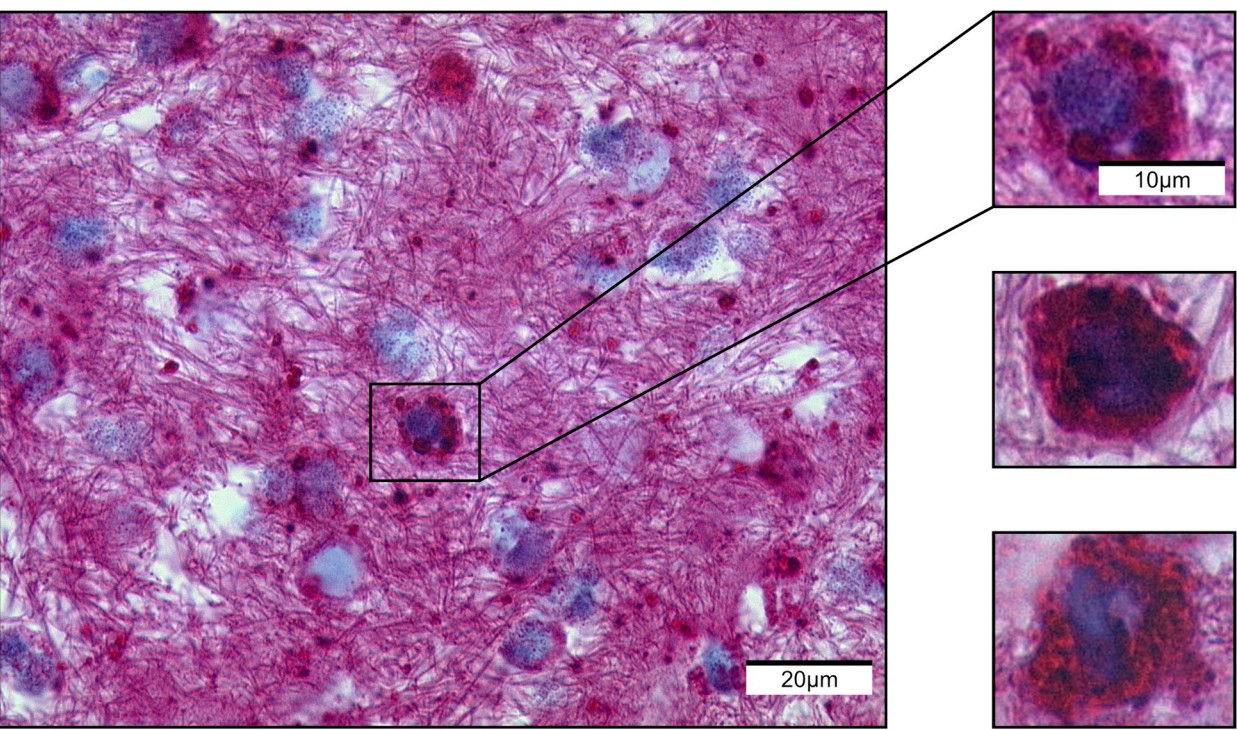

**Fig 2. Immunochemistry with mAb EmG11 of a pleural effusion.** Staining shows macrophages with intracytoplasmic spems.

germinal centers and/or sinuses (Fig 3A). In detail, we detected spems affecting the germinal center and sinuses in 35 (56%) lymph nodes of AE patients. Spems affecting solely the germinal centers were found in 27 (43%) lymph nodes of AE patients. One (2%) lymph node was affected exclusively in the sinuses. In case of CE, in most lymph nodes spegs were detected in both germinal centers and sinuses (31/39; 79%). Seven (18%) lymph nodes of CE patients showed exclusively spegs in the germinal centers. In one case (3%), we found spegs only in the sinuses.

In the germinal centers of the lymph nodes, staining revealed involvement of either the whole germinal center or the light zone only. By double immunofluorescence microscopy of one CE lymph node, using CD23 as marker for follicular dendritic cells (FDC) and mAbEmG3 for spegs, we detected a co-localization of FDCs and spegs in the germinal centers (Fig 4; Isotype control: S3 Fig). The T cell zone was not involved. In contrast to IHC, no particles like spems and spegs in lymph nodes were seen by HE and PAS staining (Fig 3B).

The lymph node measurements are shown in Table 3 and Fig 3C. The data for each individual lymph node is found in S3–S5 Tables. Data broken down by individual patient can be found in the S6 Table.

The largest mean size was measured in lymph nodes with spems of AE patients. These lymph nodes differed significantly in size from lymph nodes of AE patients without detectable spems in the lymph node (p = 0.0178). Further, lymph nodes of both groups were significantly larger than the lymph nodes of the abdominal control group (with spems: p = 0.0002; without spems: p = 0.0201). In CE, abdominal lymph nodes with spegs were larger in diameter compared to the abdominal control lymph nodes (p = 0.0003). The thoracic lymph nodes of CE patients with spegs showed no significant differences in size compared to the thoracic control group (p = 0.2259). In CE, lymph nodes without detectable spegs were smaller than lymph nodes with spegs; however, due to the small number (n = 2) further statistical analysis was not performed.

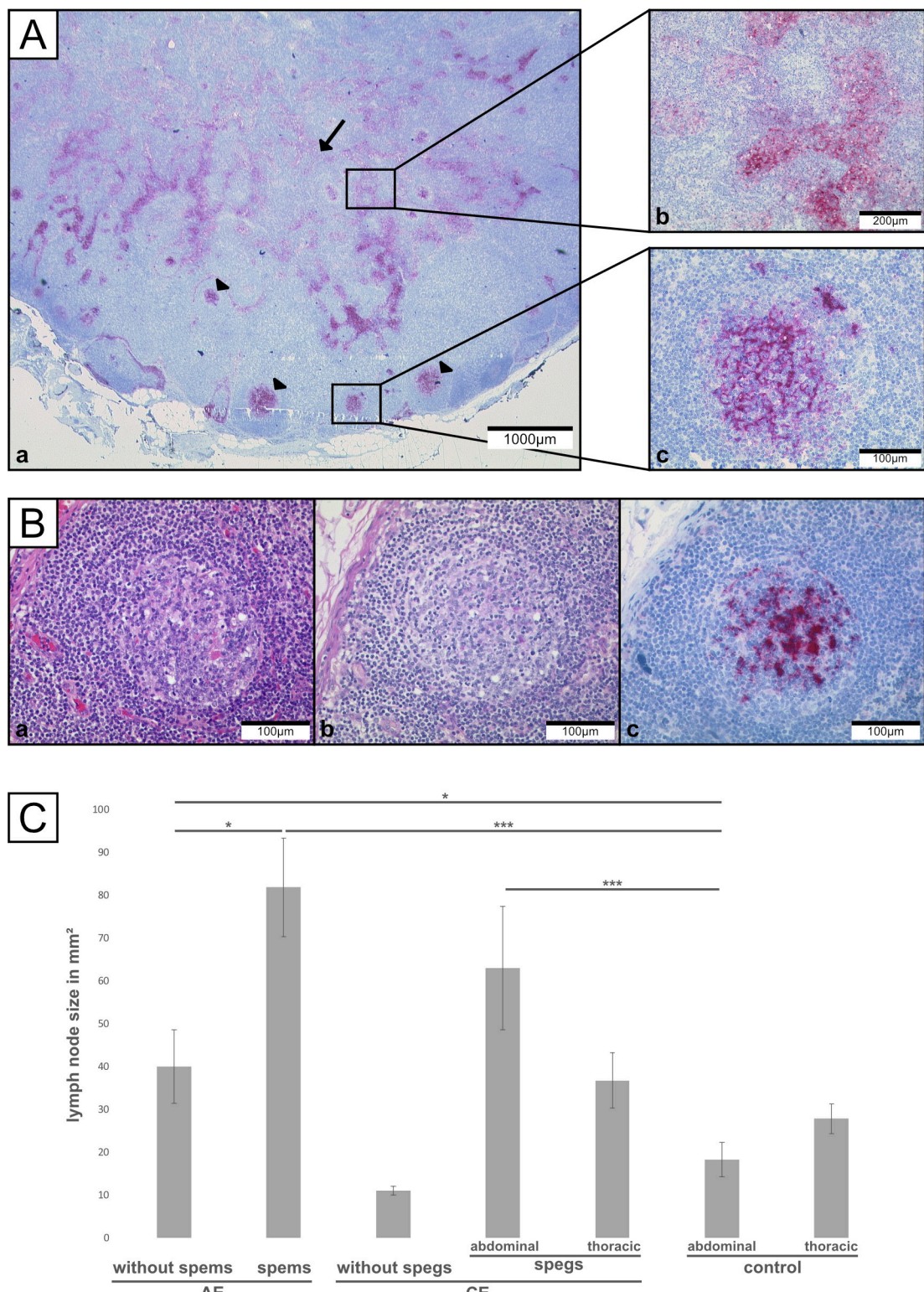

**Fig 3. Regional lymph nodes of patients with alveolar or cystic echinococcosis.** (Aa) Staining of mAb EmG3 of lymph node of a CE patient. Positive staining of spegs in germinal centers (arrowhead) and sinuses (arrow). (Ab) Higher magnification of sinus. (Ac) Higher magnification of germinal center. (B) High magnification of a germinal center (serial sections: a: HE staining; b: PAS staining; c: mAb EmG3-staining). (C) Results of size measurements of lymph nodes. (*) p<0.05 (**) p<0.01 (***) p<0.001.

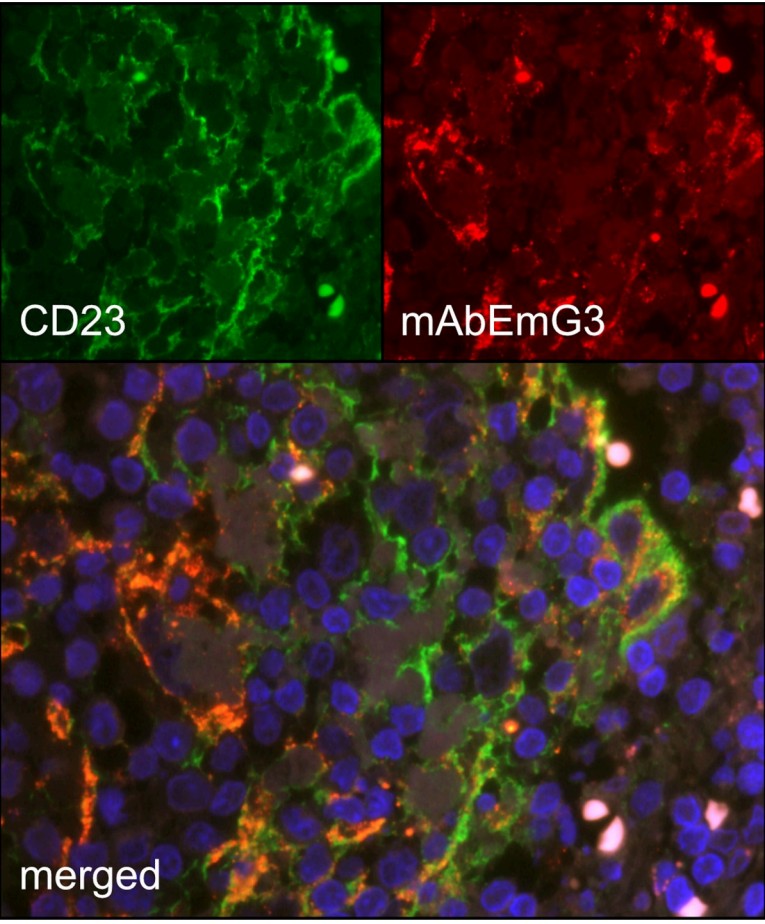

**Fig 4. Fluorescence staining of lymph node containing spegs from patients with cystic echinococcosis.** Double-immunofluorescence staining with CD23 (A; green) and mAb EmG3 (B; red) of a lymph node tissue of a patient with CE. The two antigens are co-located (C; merged, orange).

We combined data of the lymph node size and the detailed distribution of spems and spegs. Lymph nodes of AE patients with spems in the germinal center and the sinuses were larger than lymph nodes with spems in the germinal centers only. The difference was significant ($p = 0.0196$). In CE, such a correlation between affected area and size in lymph nodes was not found. Furthermore, lymph nodes with a strong (+) mAb EmG3-staining were larger than lymph nodes with less intensive staining ((+)). In AE, this was a significant difference ($p = 0.0011$). A correlation between WHO-PNM or serological data and lymph node size / EmG3 staining was not detected.

## Discussion

The histological diagnosis of AE and CE in human tissues based on routine stainings such as hematoxylin-eosin and PAS stainings is challenging for pathologists due to the rareness of this disease and may cause diagnostic problems even for experienced pathologists [9,14]. There-fore, immunohistological confirmation of diagnosis is appreciated. In this respect, the mono-clonal antibody EmG3 is an improvement in histology based diagnosis of echinococcosis. We tested the antibody in IHC on a large number of specimens and detected specific stainings for CE and AE. One main feature of the antibody is a more sensitive staining of fragments of the

**Table 3. Results of regional lymph node measurements of patients with alveolar or cystic echinococcosis and of control group without echinococcosis.**

|  | n | area (mm$^2$) | p-value |
|---|---|---|---|
| **AE (affected with spems)** | 63 | 91.75±91.43 | 0.0178[2], 0.0002[3] |
| intensity score |  |  |  |
| + (strong positive) | 24 | 128.17 ± 105.48 | 0.0011 |
| (+) (weak positive) | 39 | 53.18 ± 68.59 |  |
| localization |  |  |  |
| germinal center and sinus | 35 | 106.61 ± 99.15 | 0.0196 |
| only germinal center | 27 | 52.39 ± 71.53 |  |
| only sinus | 1 | 4 |  |
| **AE (without spems)** | 32 | 39.95 ± 48.83 | 0.0201[3] |
| **CE (affected with spegs)** | 39 | 48.80 ± 48.08 | 0.0003[4] |
| **abdominal** | 18 | 62.93 ± 61.26 | 0.0003[3] |
| **thoracic** | 21 | 36.69 ± 29.48 | 0.2259[5] |
| intensity score |  |  |  |
| + (strong positive) | 26 | 58.64± 55.49 | 0.0700 |
| (+) (weak positive) | 13 | 29.12±16.61 |  |
| localization |  |  |  |
| germinal center and sinus | 31 | 49.12 ± 49.89 | 0.9543 |
| only germinal center | 7 | 47.93 ± 46.87 |  |
| only sinus | 1 | 45 |  |
| **CE (without spegs)** | 2 | 11 ± 1.41 | n/a |
| **control (abdomen + thorax)** | 73 | 23.71±22.19 |  |
| abdominal control | 31 | 18.23 ± 13.77 |  |
| thoracic control | 42 | 27.76 ± 26.21 |  |

[1]p-values refer to the next line (if not further specified)

[2]vs. AE (without spems)

[3]vs. abdominal control group

[4]vs. abdominal and thoracic control group

[5]vs. thoracic control group; n/a: not available.

larval stage of *E. multilocularis* in comparison to mAb Em2G11. Furthermore, we confirm that mAb EmG3 is suitable as a screening tool to detect both species of *Echinococcus spp*. For additional discrimination of AE and CE, the use of the mAb Em2G11 as a specific tool for AE diagnostic is recommended [9]. We show that the diagnosis of AE and CE is possible on bioptic material without intact laminated layer, including aspirates, as EmG3 detects fragments of the laminated layer of echinococcosis. The power of this immunohistochemical approach is further shown by the fact that we conclusively confirmed the diagnosis of CE in 8 yet uncertain cases. In addition to conventional histology, PCR is assumed to be the gold standard in diagnosis of echinococcosis. Reinehr et al. showed that combined immunohistochemical staining with mAb Em2G11 and mAb EmG3 is as sensitive and specific as PCR. Compared to PCR testing, immunostaining is faster and less expensive. Moreover, immunohistochemistry is available in many laboratories around the world, which have not the possibility to perform PCR [9]. CE is a worldwide burden, with endemic areas also in rural regions and locations with no possibility for PCR techniques [2]. Patients in these areas could benefit from the immunohistological diagnostic process using the mAb EmG3. Furthermore, in our daily routine diagnostics we have observed several cases of echinococcosis with detectable spems or spegs, which were

negative by PCR. Reinehr et al. described spegs in specimens of patients with CE for the first time [9]. In addition, we detected spegs in lymph nodes of CE patients and confirmed published data regarding spems in AE [13,14,17,19]. Similar to spems, spegs are most likely generated by shedding of laminated layer material during the growth of the metacestode [14,18]. By staining with the mAb EmG3, we show that more lymph nodes in AE are affected with spems than previously described by staining with mAb Em2G11. In contrast to AE, we detected spegs in the large majority of all analyzed lymph nodes of CE patients (66.3% *versus* 95.1%). This difference was significant. Therefore, the structural differences of both cystic lesions may account for the more frequent detection of spegs in lymph nodes of patients with CE than spems in lymph nodes of patients with AE. In CE, one distinguishing feature is a thicker laminated layer than in AE [9]. Furthermore, the incubation time of CE is shorter in comparison to AE [7]. The lesion in CE may grow faster than in AE. Thus, we hypothesize that CE lesions may show higher shedding in the periphery with increased draining to the regional lymph nodes.

To date, the biological significance of spems and of spegs is not well understood. It is a common feature of helminths to inhibit the expression of proinflammatory cytokines or costimulatory molecules by various parasitic factors [24–26]. Several so called excretory/secretory (E/S)-products from the oncosphere and metacestode of *E. multilocularis* have been shown to suppress the host immune system [27]. Furthermore, E/S-products may induce apoptosis of host dendritic cells (DCs) [27]. In turn, apoptosis triggered by parasites, including viral or bacterial infections, may result in immunosuppression [28]. Previous studies pointed to such an effect of particles originated from laminated layer of *E. granulosus*. These particles have been shown to induce a stage of DCs, defined as "semi matured" DCs [29]. These DCs are supposed to reflect an intermediate stage of immature and mature DCs and are characterized by a lack of costimulatory molecules, such as CD80 and CD86, or lower levels of immunostimulatory cytokines, such as IL-12 and IL-10 [30]. This phenotype is associated with a higher tolerogenic potential comparable to immature DCs [31]. In line with this finding, proliferation of macrophages in response to IL-4 and M-CSF is inhibited by these particles [32]. For *E. multilocularis*, immunomodulatory effects of the Em2 antigen are well known. Em2 originates in the laminated layer and is also detected in spems. The IgG response to Em2 is independent of $\alpha\beta^+CD4^+T$ cells[33]. Further, no maturation of DCs is induced by Em2 [34]. These data regarding on the interaction of the immune response and antigens of the CE or AE laminated layer *in vitro* and in mice points to an important role of spems and spegs in the complex interaction of the parasite with the host. Hence, spems and spegs may influence the immune response in this direction and thereby support growth of the parasite.

The accumulation of these particles in the germinal centers and in the sinuses of lymph nodes indicate a close interaction between spems and spegs and the immune system. The accumulation of these particles in the germinal center may play a role in the B cell activation [35]. To further evaluate the significance of spegs in the germinal centers, we performed a double staining of CD23 and mAb EmG3. CD23 (FcεRII) is a low affinity Fc receptor for IgE which is expressed on FDCs in the light zone of GCs [36,37]. The staining revealed a close association of FDCs and spegs. Various functions of FDCs are described, such as the organization of the lymphoid microarchitecture [36,38], antigen capturing and antigen presentation to B cells [36,38,39], promotion of debris removal by crosslinking of apoptotic cells [40] and prevention of autoimmunity [36]. Most probably, spegs are bound by antibodies, complement factors or both, and therefore may form an immune complex [36]. After being captured by the FDCs, the immune complex including spegs can be presented to B cells which undergo somatic mutation, positive and negative selection, isotype switching, and differentiation into plasma cells or memory B cells [41]. In this scenario, spegs may serve as antigens of the parasite involved for the specific immune response of the host.

These previously described immunosuppressive effects are in contrast to the shown enlargement of the spems and spegs-affected abdominal lymph nodes and the described close interaction of spegs with FDCs. Our findings point to an increased immune response due to spems and spegs that originated from the laminated layer. Moreover, the enlargement of infiltrated lymph nodes has been described in AE, although in lower numbers [17]. In the comparison of the abdominal and the thoracic lymph nodes in CE, we observed a significant enlargement of the lymph nodes only in the abdomen. The underlying causes of this difference remains to be investigated. Our data further suggest a correlation between the severity of the infection in terms of detectable spems and spegs and the enlargement of the lymph node. A higher burden of spems was associated with an increase in size and therefore may reflect an enhanced immunogenic effect in direct correlation with the amount of spems and spegs detected.

In addition to spems and spegs in the germinal centers, we noticed these particles in the sinuses of lymph nodes. This may point to an involvement of sinus histiocytes [35]. Therefore, we hypothesize that spems and spegs drain to the regional lymph nodes *via* lymphatic fluid, reach the sinus and consequently also the germinal centers [42]. Moreover, these particles may be taken up by macrophages since we have shown that spems are present in the cytoplasm of macrophages in pleural effusions.

Since more than one lymph node was sampled within individuals, the characteristics of lymph nodes might be more similar within individuals than between individuals, and a proper statistical approach would have to consider this effect. However, due to a limited number of lymph nodes, the statistical analysis is restricted, and we could not provide valid data of the influence of individual differences on lymph node size. In conclusion, the mAb EmG3 is a highly valuable tool for the histological diagnosis of echinococcosis. The detection of spegs in lymph nodes points to a larger interface of interaction between parasite and host in CE than originally assumed. The close proximity of spems and spegs in lymph nodes with antigen presenting cells sheds new light on the immune reaction of the host to the parasite. Immunomodulating effects of fragments of the laminated layer in AE and CE are well described; based on our observation our analyses we suggest that spems and spegs lead to enlargement of the immune-reactive lymph nodes and can thereby influence the immune response.

## Supporting information

**S1 Table. Negative controls for mAb EmG3.**
(DOCX)

**S2 Table. IgM-isotype controls for mAb EmG3.**
(DOCX)

**S3 Table. Data of lymph nodes (n = 95) of 25 patients with alveolar echinococcosis.**
(DOCX)

**S4 Table. Data of lymph nodes (n = 41) of 19 patients with cystic echinococcosis.**
(DOCX)

**S5 Table. Data of control lymph nodes (n = 74) of 8 uninfected patients.**
(DOCX)

**S6 Table. Data of lymph nodes per patient.**
(DOCX)

**S1 Fig. Examples for mAb EmG3-staining intensity in lymph nodes on the examples of patients with alveolar echinococcosis.** (-) no staining in both magnifications. ((+)) weak

positive staining in 100x magnification; (+) strong positive staining in both magnifications.
(TIF)

**S2 Fig. Negative controls for mAb EmG3-stainings on tissue of patients without echinococcosis.** (left) mAb EmG3-IHC shows no specific staining. (right) Control staining without primary antibody. Only weak background staining is seen in hepatocytes and gall bladder epithelia.
(TIF)

**S3 Fig. Fluorescence control staining of CE lymph.** Double-immunofluorescence staining with CD23 (green) and IgM-isotype control (red) of a CE lymph node. No staining in isotype control.
(TIF)

## Acknowledgments

We are grateful to Aurelia Bauer (University of Ulm) for supporting us in language editing.

## Author Contributions

**Conceptualization:** Johannes Grimm, Michael Reinehr, Achim Weber, Peter Deplazes, Annika Beck, Thomas F. E. Barth.

**Data curation:** Johannes Grimm, Annika Beck, Thomas F. E. Barth.

**Formal analysis:** Johannes Grimm, Julian Schmidberger, Annika Beck, Thomas F. E. Barth.

**Funding acquisition:** Peter Möller, Thomas F. E. Barth.

**Investigation:** Johannes Grimm, Juliane Nell, Annika Beck, Thomas F. E. Barth.

**Methodology:** Johannes Grimm, Annika Beck, Thomas F. E. Barth.

**Project administration:** Johannes Grimm, Annika Beck, Thomas F. E. Barth.

**Resources:** Andreas Hillenbrand, Doris Henne-Bruns, Beate Gruener, Peter Möller, Annika Beck, Thomas F. E. Barth.

**Supervision:** Peter Deplazes, Peter Möller, Annika Beck, Thomas F. E. Barth.

**Validation:** Johannes Grimm, Annika Beck, Thomas F. E. Barth.

**Visualization:** Johannes Grimm, Annika Beck, Thomas F. E. Barth.

**Writing – original draft:** Johannes Grimm, Annika Beck, Thomas F. E. Barth.

**Writing – review & editing:** Johannes Grimm, Juliane Nell, Andreas Hillenbrand, Doris Henne-Bruns, Julian Schmidberger, Wolfgang Kratzer, Beate Gruener, Tilmann Graeter, Michael Reinehr, Achim Weber, Peter Deplazes, Peter Möller, Annika Beck, Thomas F. E. Barth.

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
