## [Decision Letter · Decision Letter 0]

14 Jul 2020

Dear Prof. Dr. Barth,

Thank you very much for submitting your manuscript "Immunohistological Detection of Small Particles of Echinococcus multilocularis and Echinococcus granulosus in Lymph Nodes is Associated with Enlarged Lymph Nodes in Alveolar and Cystic Echinococcosis" for consideration at PLOS Neglected Tropical Diseases. As with all papers reviewed by the journal, your manuscript was reviewed by members of the editorial board and by several independent reviewers. In light of the reviews (below this email), we would like to invite the resubmission of a significantly-revised version that takes into account the reviewers' comments. 

We cannot make any decision about publication until we have seen the revised manuscript and your response to the reviewers' comments. Your revised manuscript is also likely to be sent to reviewers for further evaluation.

Sincerely,

Guilherme L Werneck

Deputy Editor

Mar Siles-Lucas

Deputy Editor

Editor's comments:

The manuscript brings interesting and new information. Reviewers raised important concerns that should be considered when submitting a revised version of the manuscript. There are some extra points that authors should also consider in their revision:

1) The authors used more than one sample for each patient. Two issues need to be considered:

- Please provide information on the distribution of samples per patient (mean, SD, median, interquartile range, maximum, and the minimum number of samples per patient). These data might come in Table 2 or a supplementary table.

- Most analyses are performed at the lymph node level (e. g., comparing sizes), but this approach has an inherent statistical problem since lymph nodes are clustered within individuals. In this situation, the characteristics of lymph nodes might be more similar within individuals than between individuals. A more appropriate statistical approach taking care of this random effect should have been considered using, for instance, a mixed-effects ANOVA for comparing means between groups. 

2) Please check Table 3 and confirm that among AE (affected with spems; n=63), the numbers of strong positive and weak negative are correct (they sum up 50). If yes, please provide a legend for the other 13 samples that were not strong or weak positive.

Please note that reviewer #1 sent comments in a separate file.

Reviewer's Responses to Questions

**Key Review Criteria Required for Acceptance?**

**Methods**

-Are the objectives of the study clearly articulated with a clear testable hypothesis stated?

-Is the study design appropriate to address the stated objectives?

-Is the population clearly described and appropriate for the hypothesis being tested?

-Is the sample size sufficient to ensure adequate power to address the hypothesis being tested?

-Were correct statistical analysis used to support conclusions?

-Are there concerns about ethical or regulatory requirements being met?

Reviewer #1: (No Response)

Reviewer #2: Please see Summary and General Comments.

Reviewer #3: The work is original, well designed and the methodology used is adequate.

**Results**

-Does the analysis presented match the analysis plan?

-Are the results clearly and completely presented?

-Are the figures (Tables, Images) of sufficient quality for clarity?

Reviewer #1: (No Response)

Reviewer #2: Please see Summary and General Comments.

Reviewer #3: The results are clearly presented.

The quality of figures is adecuate.

**Conclusions**

-Are the conclusions supported by the data presented?

-Are the limitations of analysis clearly described?

-Do the authors discuss how these data can be helpful to advance our understanding of the topic under study?

-Is public health relevance addressed?

Reviewer #1: (No Response)

Reviewer #2: Please see Summary and General Comments.

Reviewer #3: The conclusions are supported by the data.

**Editorial and Data Presentation Modifications?**

Reviewer #1: (No Response)

Reviewer #2: Please see Summary and General Comments.

Reviewer #3: (No Response)

**Summary and General Comments**

Reviewer #1: (No Response)

Reviewer #2: Immunohistological Detection of Small Particles of Echinococcus multilocularis and Echinococcus granulosus in Lymph Nodes is Associated with Enlarged Lymph Nodes in Alveolar and Cystic Echinococcosis

The study described Echinococcus Small Particles in two sets of 156 AE and 77 CE as well as 8 suspected patients. The manuscript present valuable information on the immunohistochemical characteristics, host-parasite interaction and immunomodulating effects of Echinococcosis in humans using specific monoclonal antibodies (mAB). In my view the study warrants publication, however several issues need to be addressed before.

- Ref to the significance of the study, the authors failed to make a compelling case for the clinical and diagnostic implications and/or advantages of using mAb-based diagnosis. No comparisons have been made with FFPE-based molecular techniques which seem to be less expensive and more reliable. As the authors pointed out IHC with the mAbs could be an adjunct for the final histological diagnosis, in addition to the routine H&E staining that poses high diagnostic value. This is particularly true for CE.

- Did you genotype the parasites? If yes, did you find any evidence of variation within different species/genotype of E. granulosus sensu lato?

- No comparisons have been made with the initial clinical diagnosis of the patients. Were there any patients with an initial diagnosis of CE that is proved to be AE by mAb-IHC, and vice versa? This is specially important in the countries like China, Turkey and Iran where both CE and AE are co-endemic. 

- The authors use WHO-PNM classification for AE, however no detailed data were provided for WHO-IWGE classification of CE lesions.

- About 20% of AE samples originated from the lymph nodes. Does it mean 25% of AE patients have AE cystic lesions in their lymph nodes? The same is more or less true about CE, i.e. 10.7% of samples and 15.6% of CE patients. These are odd figures. Were they primary or metastatic lesions?

- Figures 1 and 3 resolutions are not adequate to support findings of the study.

- line 215: "IHC with mAb EmG3 revealed positive staining of fragments of the laminated layer and hence diagnosis of CE was confirmed." As EmG3 is genus-specific and reacts with both AE and CE, how did you reach a CE diagnosis rather than AE?

- line 232: "In 39 of 41 (95.1%) lymph nodes of CE patients we detected spegs." Please provide the number of positive patients as well (? out of 12 patients?).

- The authors found spegs and traces of laminated layer in 95% of lymph nodes of CE patients with lymph node involvements. Were there any patients with organ involvements other than lymph nodes, positive for spegs in the lymph nodes? Same comment for spems.

- No statistical analysis and P-values were provided in Table 3.

- The authors could discuss the value of SEM images in this study and recommendations for future studies.

Minor comments

- line 180: exact Fisher test > Fisher exact test 

- line 236: Protoscoleces > protoscoleces

- line 243: 7 > Seven

- line 309: may growth > may grow

Reviewer #3: The authors assessed the mAb EmG3 in a diagnostic setting using infected human tissue samples with AE and CE. Moreover, they evaluated the localization and distribution of small particles in human tissue. 

The work is original, well designed and the methodology used is adequate. Moreover, it represents an important progress for a better understanding of the role of the immune response during alveolar and cystic echinococcosis infection.

Some minor comments:

1) Author Summary. Line 30. echinococcosis with lower case.

2) Scientific names should be abbreviated after its first appearance on the text. Revise the entire manuscript. Example, line 34: change "Echinococcus granulosus s.l." to "E. granulosus s.l.".

3) Lines 90 and 91. "Cystic formations". Revise the use of the word "cyst" for alveolar echinococcosis. "Cyst" should never be used to designate the central necrotic cavity often developed in the AE lesions. “Pseudocyst” should be used for this cavity. See: International consensus on terminology to be used in the field of echinococcoses. Vuitton et al. 2020 (Parasite 27, 41).

4) Line 113. If the authors underline the first letter to show the acronym of "spegs", please do the same in line 109 for "spems".

5) Line 236. protoscoleces in lower case.

PLOS authors have the option to publish the peer review history of their article (what does this mean?). If published, this will include your full peer review and any attached files.

Reviewer #1: Yes: Hamza Avcioglu

Reviewer #2: Yes: Majid Fasihi Harandi

Reviewer #3: Yes: María Celina Elissondo
---

## [Decision Letter · Decision Letter 1]

30 Sep 2020

Dear Prof. Dr. Barth,

Thank you very much for submitting your manuscript "Immunohistological Detection of Small Particles of Echinococcus multilocularis and Echinococcus granulosus in Lymph Nodes is Associated with Enlarged Lymph Nodes in Alveolar and Cystic Echinococcosis" for consideration at PLOS Neglected Tropical Diseases. As with all papers reviewed by the journal, your manuscript was reviewed by members of the editorial board and by several independent reviewers. The reviewers appreciated the attention to an important topic. Based on the reviews, we are likely to accept this manuscript for publication, providing that you modify the manuscript according to the review recommendations. 

All the comments have been considered and adequately addressed, but still there are two minor questions that should be revised:

1) Editor, ref to the comment #2: I suggest changing the phrase included (“Due to a limited number of lymph nodes the statistical analysis is restricted. Therefore, we cannot provide valid data of the influence of individual differences of lymph node size.”) by the following ("Since more than one lymph node was sampled within individuals, the characteristics of lymph nodes might be more similar within individuals than between individuals, and a proper statistical approach would have to consider this effect. However, due to a limited number of lymph nodes, the statistical analysis is restricted, and we could not provide valid data of the influence of individual differences on lymph node size.”)

2) Reviewer #2, ref to the comment #4: please add statements in the manuscript regarding this limitation.

Sincerely,

Guilherme L Werneck

Deputy Editor

Mar Siles-Lucas

Deputy Editor

All the comments have been considered and adequately addressed, but still there are two minor questions that should be revised:

1) Editor, ref to the comment #2: I suggest changing the phrase included (“Due to a limited number of lymph nodes the statistical analysis is restricted. Therefore, we cannot provide valid data of the influence of individual differences of lymph node size.”) by the following ("Since more than one lymph node was sampled within individuals, the characteristics of lymph nodes might be more similar within individuals than between individuals, and a proper statistical approach would have to consider this effect. However, due to a limited number of lymph nodes, the statistical analysis is restricted, and we could not provide valid data of the influence of individual differences on lymph node size.”)

2) Reviewer #2, ref to the comment #4: please add statements in the manuscript regarding this limitation.

Reviewer's Responses to Questions

**Key Review Criteria Required for Acceptance?**

**Methods**

-Are the objectives of the study clearly articulated with a clear testable hypothesis stated?

-Is the study design appropriate to address the stated objectives?

-Is the population clearly described and appropriate for the hypothesis being tested?

-Is the sample size sufficient to ensure adequate power to address the hypothesis being tested?

-Were correct statistical analysis used to support conclusions?

-Are there concerns about ethical or regulatory requirements being met?

Reviewer #2: Yes

Reviewer #3: Ok

**Results**

-Does the analysis presented match the analysis plan?

-Are the results clearly and completely presented?

-Are the figures (Tables, Images) of sufficient quality for clarity?

Reviewer #2: Yes

Reviewer #3: OK

**Conclusions**

-Are the conclusions supported by the data presented?

-Are the limitations of analysis clearly described?

-Do the authors discuss how these data can be helpful to advance our understanding of the topic under study?

-Is public health relevance addressed?

Reviewer #2: Yes

Reviewer #3: OK

**Editorial and Data Presentation Modifications?**

Reviewer #2: All the comments have been considered and properly addressed. Ref to the comment #4 please add statements in the manuscript regarding this limitation.

Reviewer #3: (No Response)

**Summary and General Comments**

Reviewer #2: All the comments have been considered and properly addressed. Ref to the comment #4 please add statements in the manuscript regarding this limitation.

Reviewer #3: (No Response)

PLOS authors have the option to publish the peer review history of their article (what does this mean?). If published, this will include your full peer review and any attached files.

Reviewer #2: No

Reviewer #3: No
---

## [Editor Report · Decision Letter 2]

26 Oct 2020

Dear Prof. Dr. Barth,

We are pleased to inform you that your manuscript 'Immunohistological Detection of Small Particles of Echinococcus multilocularis and Echinococcus granulosus in Lymph Nodes is Associated with Enlarged Lymph Nodes in Alveolar and Cystic Echinococcosis' has been provisionally accepted for publication in PLOS Neglected Tropical Diseases.

Best regards,

Guilherme L Werneck

Deputy Editor

Mar Siles-Lucas

Deputy Editor

---

## [Editor Report · Acceptance letter]

3 Dec 2020

Dear Prof. Dr. Barth,

We are delighted to inform you that your manuscript, "Immunohistological Detection of Small Particles of Echinococcus multilocularis and Echinococcus granulosus in Lymph Nodes is Associated with Enlarged Lymph Nodes in Alveolar and Cystic Echinococcosis," has been formally accepted for publication in PLOS Neglected Tropical Diseases.

Best regards,

Shaden Kamhawi

co-Editor-in-Chief

Paul Brindley

co-Editor-in-Chief
